# Applicability of a Combined DAF-MF Process to Respond to Changes in Reservoir Water Quality through a Two-Year Pilot Plant Operation

**DOI:** 10.3390/membranes11120964

**Published:** 2021-12-07

**Authors:** Joon-seok Kang, Jayeong Seong, Jewan Yoo, Pooreum Kim, Kitae Park, Jaekyu Lee, Jihoon Cheon, Hyungsoo Kim, Sangyoup Lee

**Affiliations:** 1Graduate School of Water Resources, Sungkyunkwan University, Suwon 16149, Korea; joonseok0724@gmail.com (J.-s.K.); sujayou@skku.edu (J.S.); yjw19929@naver.com (J.Y.); poorerm@naver.com (P.K.); kiotae@skku.edu (K.P.); 2Toray Advanced Materials Korea Inc., Seoul 07790, Korea; jaekyu.lee@torayamk.com (J.L.); jihoon.cheon@torayamk.com (J.C.)

**Keywords:** dissolved air flotation (DAF), microfiltration (MF), DAF-MF combined process, high-turbidity raw water, membrane fouling

## Abstract

The optimal operating conditions of a combined dissolved air flotation (DAF)-microfiltration (MF) process to respond to changes in raw water quality were investigated by operating a pilot plant for two years. Without DAF pre-treatment (i.e., MF alone), MF operated stably with a transmembrane pressure (TMP) increase of 0.24 kPa/d when the turbidity of raw water was low and stable (max. 13.4 NTU). However, as the raw water quality deteriorated (max. 76.9 NTU), the rate of TMP increase reached 43.5 kPa/d. When DAF pre-treatment was applied (i.e., the combined DAF-MF process), the MF process operated somewhat stably; however, the rate of TMP increase was relatively high (i.e., 0.64 kPa/d). Residual coagulants and small flocs were not efficiently separated by the DAF process, exacerbating membrane fouling. Based on the particle count analysis of the DAF effluent, the DAF process was optimised based on the coagulant dose and hydraulic loading rate. After optimisation, the rate of TMP increase for the MF process stabilised at 0.17 kPa/d. This study demonstrates that the combined DAF-MF process responded well to substantial changes in raw water quality. In addition, it was suggested that the DAF process must be optimised to avoid excessive membrane fouling.

## 1. Introduction

As water contamination intensifies along with rapid industrial development and climate change due to global warming, numerous studies are being conducted to develop sustainable technologies for producing safe drinking water from contaminated source water with high turbidity, trace organic matter, toxic inorganic compounds, and colour/odour-causing materials [1,2]. Drinking water treatment plants in Korea generally use several unit processes, such as coagulation/flocculation, sedimentation, and sand filtration, followed by chlorine disinfection. The amounts of coagulants and chlorine used in the pre-treatment process to improve the sand filtration efficiency are high to ensure water safety, which has resulted in the occurrence of residual coagulants and disinfection by-products (DBPs) in treated water [3]. To resolve this problem, an advanced membrane filtration process has been introduced.

Membrane filtration can remove target contaminants well and has a smaller footprint and easier maintenance than existing conventional processes [4]. Membrane filtration processes are also rapidly being introduced for replacing rapid sand filtration due to their numerous advantages, such as simple equipment, automated operation, and water quality improvement [5]. The number of domestic membrane water treatment plants has been rapidly increasing since 2009, beginning with the introduction of the EDONG membrane water treatment plant in Yangpyeong-Gun in 2014 [6,7]. As of 2017, 27 domestic membrane filtration plants have been introduced, with a total plant capacity of 442,080 m^3^/d [6,7].

In the membrane filtration process, it is important to ensure a sufficient quantity and quality of the produced water while minimising membrane fouling. The most common type of fouling in microfiltration (MF) and ultrafiltration (UF) is the accumulation of particles, colloids, and organic matter on the membrane’s surface, forming a cake layer that impedes water permeation, consequently increasing the operating pressure and shortening the duration of stable operation [8]. To minimise membrane fouling, the pre-treatment processes of coagulation/flocculation followed by sedimentation could be employed prior to membrane filtration. However, to apply these pre-treatment processes, additional costs for area, installation, and coagulant chemicals are incurred, depending on the source water quality as well as the type of pre-treatment [9,10].

When coagulants are added during the pre-treatment process, they uniformly diffuse within a short time by hydrolysis to neutralise and coagulate colloidal particles in the raw water and form easily settleable large flocs. Therefore, the efficiency of the membrane process can be increased by reducing membrane fouling and improving the process performance by implementing sedimentation after coagulation/flocculation as a pre-treatment process [11]. The most common coagulants used in drinking water treatment are alum and polyaluminum chloride (PAC) [12]. PAC hydrolyses more easily than alum and emits polyhydroxides with long molecular chains and high electrical charge in the solution with a low alkali consumption, thus maximising the physical action of flocculation [13]. In addition, PAC achieves better coagulation than alum in medium- and high-turbidity water [13]. Floc formation with PAC is rapid, and the sludge produced is more compact than that produced by alum [12,13]. However, membrane fouling can be accelerated by the formation of metal salt hydrolysates if the coagulant dose and hydraulic loading are not optimised [14,15]. In addition, applying coagulation/flocculation without sedimentation transfers the flocs directly to the membrane process, forming a cake layer on the membrane surface, leading to a decrease in the flux and reducing the efficiency of the overall process.

Lakes and reservoirs are closed bodies of water that frequently experience high turbidity and eutrophication, depending on the season and weather. In addition, as green algae cause problems in taste and odour, special care is required when using lakes or reservoirs as raw water for drinking water treatment. Since 2000, the application of dissolved air flotation (DAF) as an alternative process for water purification has been increasing worldwide. The DAF process has a superior performance in removing algae and colour-causing materials and in treating high-turbidity water than existing sedimentation processes. Most of the turbidity-causing substances in raw water are colloidal substances with negatively charged surfaces, which do not easily settle due to electrostatic repulsion and are stably suspended in water [16]. The DAF process is more effective in removing algae than the conventional sedimentation process when low-density algae are generated and introduced into drinking water treatment plants [17,18,19,20]. In addition, the DAF process has a higher surface load than existing sedimentation processes, allowing it to secure sufficient treatment capacity at a low footprint; as such, its application in seawater desalination and wastewater treatment facilities is increasing [21,22,23,24].

However, there is a lack of studies in Korea on attempts of reducing membrane fouling in the downstream membrane filtration process by pre-treating these materials with DAF. Currently, there are 13 domestic membrane water treatment plants with a capacity of at least 5000 tonnes/day that use lakes and reservoirs as raw water for drinking water treatment. Until yet, the pre-treatment option of all 13 plants is a conventional sedimentation process. The design flux for domestic membrane processes ranges from 1.0 to 3.0 m^3^m^−2^d^−1^ (approximately 40 to 125 LMH). This relatively wide flux range allows for the maximum flux to maintain stable water production, even in emergencies (such as the occurrence of high turbidity, algal bloom, or manganese pollution). In normal operation, the flux is typically maintained in a range from 1.0 to 1.7 m^3^m^−2^d^−1^ (approximately 40 to 70 LMH) [25]. During normal operation, stable membrane process operation is often difficult to achieve, especially in case the raw water quality deteriorates significantly. Variations in raw water quality are also increasing due to climate change, and the need for stable membrane process operations that can respond well to changes in raw water quality is increasing.

In this study, the applicability of the combined DAF-MF process to respond to changes in raw water quality was evaluated through a pilot plant operation over two years. This study is the first in Korea to evaluate the applicability of the combined DAF-MF process according to changes in raw water quality through a long-term pilot operation and a rare case (i.e., variable operation of MF-alone and combined DAF-MF process to respond to raw water quality) in the world [26,27,28]. The combined DAF-MF process can be operated with either MF-alone or as a combined DAF-MF process depending on the quality of the raw water. The performance of the combined DAF-MF process was evaluated according to the raw water quality by varying the operating conditions (i.e., flux and backwashing cycle) of the DAF and MF processes. Chemical-enhanced backwashing (CEB) and clean-in-place (CIP) were performed as needed, and the chemical cleaning waste solutions were analysed to identify the membrane foulants. In addition, the cause of changes in the performance of the combined DAF-MF process according to the raw water quality under different operating conditions was elucidated through various analytical measurements. The optimal operating conditions for the combined DAF-MF process responded well to changes in raw water quality, and the operational factors affecting the performance of the combined DAF-MF process are discussed.

## 2. Materials and Methods

### 2.1. Raw Water Quality

Raw water from the Hoedong Reservoir was supplied to the pilot plant. The average turbidity of the raw water throughout the operation period of the combined DAF-MF process was 5.62 NTU, and the maximum turbidity during the summer and rainy season reached 83.0 NTU. The average and maximum dissolved organic carbon (DOC) concentrations were 3.06 and 4.86 mg/L, respectively. The characteristics of the raw water used in this study are summarised in Table 1, including the average, maximum, and minimum values. The average turbidity, DOC, Mn, and Chl-a removal efficiencies of the DAF process were 96%, 35%, 65%, and 100%, respectively, as shown in Table 1. In the case of the Hoedong Reservoir, there was a problem that the concentration of Mn and Chl-a increased intermittently, so these were included in the water quality analysis. Raw water was used as the feed for the MF-alone process, and the DAF effluent was used as the feed water for MF in the combined DAF-MF process.

### 2.2. DAF-MF Combined Process

A polyvinylidene fluoride (PVDF) hollow fibre membrane module (HFU-2020N, Toray Advanced Materials Korea Inc., Seoul, Korea) was installed in the MF process of the pilot plant constructed at the Hoedong Reservoir in Busan. The membrane had a nominal pore size of 0.01 μm and an effective membrane surface area of 72 m^2^. An outside-in filtration mode was adopted by mounting one module in the MF process. The other specifications of the membrane module are listed in Table 2.

A schematic of the combined DAF-MF process is shown in Figure 1. Owing to the inclusion of a bypass system, the entire process could be operated as an MF-alone or combined DAF-MF process. The MF system was capable of automatic operation with continuous and repetitive water fulling, filtration, backwashing, air blowing, and draining control. Membrane cleaning was also performed automatically according to the Toray maintenance cleaning (TMC) method, as described in Table 3. In this study, the TMC cycle was changed from once per week to once per day when the raw water turbidity exceeded 30 NTU for 3 d or more. The other filtration, backwash, air blowing, TMC, and CIP conditions are summarised in Table 3. The MF system could manage a flux up to 2.8 m^3^m^−2^d^−1^ or 117 LMH. TMP was measured continuously and recorded automatically by electronic pressure gauges installed in the feed, permeate, and circulation lines. The DAF system consisted of a chemical mixing tank and a flotation tank comprising contact and separation zones. PAC (10% aluminium as Al_2_O_3_) was used as the coagulant, and the range of optimal coagulant dose pre-determined was 40–60 mg/L in terms of turbidity removal, depending on the raw water quality. The optimal hydraulic loading rate of the DAF system ranged from 10 to 15 m/h (i.e., mostly applied in the literature [17,18,19,20]), and six nozzles for generating microbubbles were located at the bottom of the flotation tank.

### 2.3. Operation and Maintenance of MF System

The operating sequence of the MF system, depending on the raw water turbidity, is illustrated in Figure 2. During normal operation with raw water turbidity below 30 NTU, the system operated in the following sequence: fulling, filtration, backwashing, air blowing, and draining, as shown in Figure 2a, and this sequence was repeated as long as the normal operation continued. If the turbidity of the raw water exceeded 30 NTU for more than 3 d, or if the TMP of the MF system increased rapidly above a certain level, the MF system could automatically apply Toray’s drain-backwash operation (i.e., an emergency operation). As shown in Figure 2b, the drain-backwash operation has one more drain and backwashing step than the normal operation. During the drain-backwash operation, the MF system recovery (=yield) decreased from 96% to 94%, while the permeate flux was maintained.

### 2.4. Analytic Measurements

Various analyses, including particle count analysis, energy dispersive x-ray spectrometry (EDS), and Fourier-transform infrared spectroscopy (FTIR), were performed to better understand the performance of the combined DAF-MF process. The particle number and size distribution in DAF effluent samples obtained under various coagulant doses and hydraulic loading rates were determined using a Mastersizer 3000 (Malvern Panalytical Ltd., Malvern, UK). The optimal operating conditions for the DAF process were determined based on the results of particle count analysis. EDS and FTIR analyses were performed on the sediment in the raw water-collection tank to identify the source of membrane foulants, as described in a previous study [29]. EDS and FTIR analyses were performed at the Cooperative Center for Research Facilities (CCRF) at Sungkyunkwan University, Korea, using JEOL JSM-7500F (JEOL Ltd., Tokyo, Japan) and Bruker IFS-66/S and TENSOR27 (Bruker Co., Billerica, MA, USA), respectively. In addition, the turbidity, DOC, Mn, and Chl-a levels of the raw water and Al, Mn, Si, Ca, and DOC levels of the chemical cleaning waste solutions were measured according to standard methods.

### 2.5. Determination of MF Process Performance

The degree of membrane fouling was estimated by calculating the total resistance (*R_t_* (m^−1^)) of the MF system using Equation (1), where *ΔP* is the transmembrane pressure (Pa), *μ* is the dynamic viscosity (Pa s), and *J* is the permeate flux (m^3^/m^2^/s).
(1)$$R_{t}=∆P/\left(\mu J\right)$$

The rate of *TMP* increase (kPa/d) was calculated using Equation (2), which estimates the rate of membrane fouling in the MF system. *TMP_i_* is the *TMP* (kPa) at the initial time of operation, *TMP_f_* is the *TMP* (kPa) at the end of operation, and t is the operating time (day).
(2)$$TMPincreaserate=\left(TMP_{f}-TMP_{i}\right)/t$$

*TMP* recovery (%) was determined from the change in the *TMP* before and after CIP using Equation (3). The effect of each chemical cleaning solution on the removal of foulants from the membrane surface can be estimated by measuring the *TMP* recovery at each CIP step with different chemical cleaning solutions. *TMP_b_* and *TMP_a_* are the TMPs before and after CIP, respectively.
(3)$$TMPRecovery\left(\%\right)=\left(TMP_{b}-TMP_{a}\right)/\left(TMP_{b}-TMP_{i}\right)\times 100$$

## 3. Results and Discussion

### 3.1. Performance of MF-Alone Process for Low-Turbidity Raw Water

Domestic membrane water treatment plants are generally operated in the flux range of 48–70 LMH [6]. The performance of the MF-alone process was evaluated at a flux of 70 LMH using raw water from Hoedong Reservoir, and the results are shown in Figure 3. Normal operation with a one-week TMC cycle was applied during this period. The TMP was monitored for 700 h (approximately 29 d) of operation; the initial TMP was 22.3 kPa (3.23 psi), and it increased to 30.6 kPa after 700 h of operation (i.e., an 8.3 kPa increase over 29 d). In this case, the rate of increase in the TMP (Equation (2)) was approximately 0.28 kPa/d. During this period, the average raw water turbidity was 3.4 NTU (maximum 11.1 NTU), maintaining stable water quality, and the permeate turbidity was less than 0.05 NTU. As the raw water quality was suitable and stable, it was confirmed that the performance of the MF-alone process operated at a flux of 70 LMH without pre-treatment was highly stable and satisfactory.

As shown in Figure 3, the total resistance (Equation (1)) during this period was calculated to be 2.056 × 10^12^ m^−1^ (min. 1.818 × 10^12^ m^−1^ and max. 2.448 × 10^12^ m^−1^). Although the total resistance increased at regular intervals, it could be easily restored by backwashing. Periodic backwashing prevented a gradual increase in resistance so that the overall resistance did not significantly increase over the experimental period. Therefore, it can be assumed that the main cause of the increase in the overall resistance was reversible fouling during this period when normal operation with a TMC cycle of once per week was applied.

To evaluate the applicability of high flux in the MF-alone process when the raw water quality was suitable and stable, the flux was increased to 80 LMH, and the performance of the MF-alone process was evaluated. Normal operation with a once-per-week TMP cycle was also applied during this period. The MF-alone process was operated for 1500 h (approximately 63 d), and the MF performance in terms of TMP and total resistance is shown in Figure 4. The average raw water turbidity during this period was 3.3 NTU (max. 13.4 NTU), which was not significantly different from the water quality during the previous period. According to the monitoring results of the TMP during 1500 h of operation (approximately 63 d), the TMP increased from 36.1 to 51.0 kPa after 1500 h of operation (i.e., a 14.9 kPa increase over 63 d). In this case, the rate of TMP increase was approximately 0.24 kPa/d.

As shown in Figure 4, the total resistance was calculated to be 2.057 × 10^12^ m^−1^ (min. 1.755 × 10^12^ m^−1^ and max. 2.558 × 10^12^ m^−1^). Similar to the previous case, when applying a flux of 70 LMH, the total resistance increased at regular intervals; however, the increase in the total resistance was not notable. Even under a flux of 80 LMH and normal operating conditions, the stable operation was possible for all operating periods without a rapid increase in TMP for the MF-alone process. Considering these results, as shown in Figure 3 and Figure 4, when the raw water quality is suitable and stable (maximum turbidity = 13.4 NTU), stable operation of the MF-alone process could be achieved, even under the high flux condition of 80 LMH without pre-treatment.

### 3.2. Performance of MF-Alone Process for High-Turbidity Raw Water

After the MF-alone process was operated stably for approximately 92 d at fluxes of 70 LMH (Figure 3) and 80 LMH (Figure 4), the raw water turbidity increased rapidly and fluctuated significantly owing to the rainy season. The performance of the MF-alone process was then evaluated for high-turbidity raw water at a flux of 80 LMH. The TMP and total resistance were monitored for 700 h (approximately 29 d), as shown in Figure 5. The average raw water turbidity during this period was 8.3 NTU, and the maximum turbidity was 76.9 NTU. As shown in Figure 5, the TMP increased noticeably during the overall period. The initial TMP was 40.4 kPa; however, as raw water with high turbidity was fed to the MF-alone process without pre-treatment, the TMP rapidly increased to 56.7 kPa within 9 h (i.e., an increase of 16.3 kPa over 9 h). At this time, the rate of TMP increase was approximately 43.5 kPa/d, which was significantly higher than that when low-turbidity water was supplied to the MF-alone process. To mitigate this rapid increase in TMP, the drain-backwash operation was applied, rather than the normal operation, and the TMC cycle was changed from once per week to once per day. With these changes, the MF-alone process system recovery decreased from 96% to 94% while maintaining a flux of 80 LMH. These changes in the operation method and TMC cycle stabilised the rate of TMP increase to 1.41 kPa/d, enabling acceptable operation of the MF-alone process for approximately 21 d. However, as shown in Figure 5, the TMP eventually reached 70 kPa after 21 d of operation. CIP was performed for the first time after approximately 113 d of pilot plant operation. After the first CIP, the TMP was restored to 47.8 kPa; however, it rapidly increased to 116.6 kPa. The rate of TMP increase during this period was approximately 8.26 kPa/d. During this period, a large amount of sediment was found in the raw water-collection tank of the pilot plant. Sediment samples from this tank were collected, and their composition was determined by EDS and FTIR analyses. The results from EDS and FTIR analyses were included in Section 3.5.

These results indicate that supplying high-turbidity raw water directly to the membrane process without pre-treatment causes a rapid increase in the TMP. In this case, even intensive backwashing (i.e., drain-backwash) and frequent chemical-enhanced backwashing (i.e., daily TMC cycle) cannot easily mitigate the increase in TMP. In addition, after CIP, the TMP could be restored, but as raw water with high turbidity was continuously supplied, the TMP of the MF-alone process increased rapidly. During this period, the total resistance more than tripled from the initial total resistance of 1.792 × 10^12^ to 5.865 × 10^12^ m^−1^ at the end of the operation. The above results suggest that it is difficult to respond to the deterioration of raw water quality using MF alone. At the end of this period, a second CIP was performed, and the chemical cleaning waste solution obtained from the CIP was analysed to identify the membrane fouling components. The results of foulants analysis with the cleaning waste solution are presented in Section 3.5.

### 3.3. Performance of DAF-MF Combined Process for High-Turbidity Raw Water

Owing to the deteriorating and fluctuating raw water quality, the performance of the combined DAF-MF process was evaluated at a flux of 80 LMH. The average raw water turbidity during this period was 8.6 NTU, and the maximum turbidity was 81.3 NTU. The operational conditions for DAF pre-treatment were as follows: coagulant (PAC) dose of 60 mg/L and hydraulic loading rate of 15 m/h. As shown in Figure 6, the initial TMP of the MF process was 36.0 kPa, and after 1700 h (approximately 71 d) of operation, it increased by 47.1 kPa, resulting in a final TMP of 83.1 kPa. The rate of TMP increase was 0.64 kPa/d, approximately 2.7 times higher than the 0.24 kPa/d determined at the same flux of 80 LMH (Figure 4). It should be noted that the results in Figure 4 were obtained from the MF-alone operation when the raw water quality was low and stable.

Although the raw water turbidity was quite high, the rate of TMP increase of 0.64 kPa/d was rather high, considering that the process consisted of both DAF and MF. Membrane processes often suffer from severe fouling if the efficiency of the upstream pre-treatment process is not optimised to reduce membrane fouling [30]. This is because potential membrane foulants can enter the downstream membrane process if the pre-treatment efficiency is poor. Additionally, pre-treatment by-products, such as small flocs and unreacted coagulants, can enter the downstream membrane process, exacerbating membrane fouling. Therefore, the operating conditions of pre-treatment processes must be optimised to effectively reduce the membrane fouling burden. Relatively small flocs and unreacted residual coagulants accumulate in the cake layer on membrane surfaces, narrowing the pores and decreasing the porosity of the cake layer, thereby increasing the cake layer resistance to water permeation [31,32]. Therefore, to ensure that the DAF process is effective without burdening the membrane process, the optimal conditions, such as the amount of coagulant and hydraulic loading rate, must be determined.

To evaluate the effect of improperly formed small flocs and unreacted residual coagulant on the acceleration of membrane fouling, particle count analysis was conducted on the DAF effluent under different coagulant doses and hydraulic loading rates. The results of particle count analysis are shown in Figure 7. The total particle count was measured at coagulant doses of 40 and 60 mg/L, and the particle size distribution was analysed at hydraulic loading rates of 10 and 15 m/h and coagulant doses of 40 and 60 mg/L. The total particle count was the lowest when the coagulant dose was 40 mg/L, and the hydraulic loading rate was 10 m/h, at 14,644 particles/mL. These conditions are significantly different from those at a coagulant dose of 60 mg/L and hydraulic loading rate of 15 mg/h in the combined DAF-MF process, wherein the DAF process was relatively ineffective at preventing an increase in TMP during the MF process. As shown in Figure 7a, over 20,000 particles/mL were present at a coagulant dose and hydraulic loading rate of 60 mg/L and 15 m/h, respectively. Figure 7b shows that, as the coagulant dose and hydraulic loading rate increased, the number of particles smaller than 25 μm increased, with the number of particles smaller than 10 μm being the highest. Based on these results, it can be inferred that small flocs and unreacted coagulant particles generated due to the poor efficiency of the DAF process deteriorated the performance of the combined DAF-MF process. At the end of this period, the third CIP was performed, and the components of the chemical cleaning waste solution were analysed to determine the possibility of small flocs and residual flocculants entering the MF process. The results of foulants analysis with the cleaning waste solution are presented in Section 3.5.

### 3.4. Performance of DAF-MF Combined Process after Optimising DAF Operating Conditions

The MF process in the DAF-MF process was operated for approximately 1400 h (approximately 58 d) at a flux of 80 LMH, and the performance of the MF process is shown in Figure 8. This operation was also conducted in cases where the raw water quality deteriorated and fluctuated, with average raw water turbidity of 10.1 NTU and maximum turbidity of 83.0 NTU. The raw water turbidity was comparable to that during the rainy season. The coagulation conditions of the DAF process were as follows: coagulant dose of 40 mg/L and hydraulic loading rate of 10 m/h, which were the optimal coagulation conditions determined from the particle count analysis, as described in the previous section. As shown in Figure 8, the initial TMP was 30.3 kPa, and it increased to 40.2 kPa after 63 d, an increase of approximately 9.9 kPa. In addition, there was no rapid increase in the TMP during this period, as shown in Figure 8, and the combined DAF-MF process operated stably, even with high-turbidity raw water. After optimising the coagulation conditions for the DAF process, the rate of TMP increase was 0.17 kPa/d, which is significantly lower (less than one-third) than that obtained prior to optimising the DAF process (0.64 kPa/d), as shown in Figure 6.

These results strongly indicate that the operating conditions of the pre-treatment process should be optimised before applying pre-treatment for the membrane process; otherwise, this may interfere with the stable operation of the membrane process. The rate of TMP increase of 0.17 m/d was even lower than that for the cases shown in Figure 3 (i.e., rate of TMP increase = 0.28 kPa/d at 70 LMH) and Figure 4 (i.e., rate of TMP increase = 0.24 kPa/d at 80 LMH), the results of which were obtained using the MF-alone process at a suitable and stable feed water quality. Thus, when a pre-treatment process is introduced, the optimisation and efficiency of the pre-treatment process are essential not only to reduce the fouling burden of the membrane process but also to avoid the acceleration of membrane fouling, especially in the DAF-MF combined process.

### 3.5. Water Quality Analysis and CIP Results

As shown in Figure 5, when the TMP increased more rapidly after the second CIP of the MF-alone process, a large amount of sediment accumulated in the raw water-collection tank. The composition of this sediment was analysed using EDS and FTIR, and the results are presented in Figure 9. The EDS analysis results, shown in Figure 9a, indicate that large amounts of Si and Mn were detected. The Si may have originated from the SiO_2_ constituting the cell walls of diatoms and from the high-turbidity raw water, and it was assumed that a significant amount of Mn was present in the high-turbidity raw water at this time. Additionally, as shown in Figure 9b, the six peaks detected by the FTIR analysis were consistent with the IR peaks of algal organic matter (AOM) and cell wall material detected in other studies [33,34]. Therefore, it appears that the rapid increase in TMP during the MF-alone process was due to the increase in algae-related materials and the substantial content of manganese in raw water. However, despite the introduction of high-turbidity raw water with potential membrane foulants, the combined DAF-MF process operated stably without a rapid increase in the TMP, as shown in Figure 8. This confirms that the combined DAF-MF process responded well to variations in the raw water quality.

Three CIPs were performed during the two-year pilot plant operation period (Figure 5 and Figure 6). The water quality of the cleaning chemical waste solutions obtained from the second and third CIPs was analysed. The membrane foulants for the MF-alone process could be estimated by analysing the water quality of the chemical cleaning waste solutions obtained from either of the first two CIPs, as the first two CIPs were performed during the MF-alone process (Figure 5). By analysing the water quality of the chemical cleaning waste of the third CIP (Figure 6), whether membrane fouling worsened because of small-sized flocs and residual coagulants due to the inefficient operation of the DAF process in the combined DAF-MF process could be evaluated. The chemical cleaning solutions used in the CIP are listed in Table 4. The TMP recovery was determined by comparing the TMP before and after CIP (Equation (3)). The concentration of substances eluted during CIP was measured by analysing the water quality of each chemical cleaning waste solution, and the results are presented in Table 4. For the second CIP performed for the MF-alone process, the TMP recoveries achieved by the sodium hypochlorite (NaOCl) and citric acid (C_6_H_8_O_7_) cleaning solutions were 64.6% and 32.1%, respectively, being relatively high. In the NaOCl cleaning waste solution, DOC was detected at a high concentration of 112 mg/L. In the citric acid cleaning waste solution, Mn was detected at a high concentration of 147 mg/L. Therefore, as shown in Figure 5, the marked increase in the TMP during the MF-alone process can be attributed to membrane fouling because of AOM and Mn. This is in suitable agreement with the results of the EDS and FTIR analyses (Figure 9).

Before optimising the DAF operating conditions, water quality analysis was conducted using the chemical cleaning waste solutions obtained from the third CIP (Figure 6). Both NaOCl (TMP recovery = 39.7%) and sulfuric acid (H_2_SO_4_) (TMP recovery = 36.4% and 23.9%) cleaning achieved reasonable TMP recovery after the third CIP, as shown in Table 4. On analysing the chemical cleaning waste obtained from the third CIP, Al and DOC were detected at very high concentrations of 45 and 91 mg/L, respectively. During this period, the average Al concentration of raw water was 0.02 mg/L; therefore, the 45 mg/L of Al in the NaOCl cleaning waste solution was considered to be derived from unreacted coagulants. Additionally, the average DOC concentration of the DAF effluent in this period was 1.8 mg/L (i.e., DOC removal was approximately 35%), while the DOC concentration measured in the NaOCl cleaning waste solution was 91 mg/L. These results indicate that fine organic flocs and unreacted coagulants not properly separated by the DAF process contributed to the increase in TMP during MF in the combined DAF-MF process (Figure 6). Therefore, it is important to optimise the operating conditions of the DAF process to ensure that flocs and residual coagulants are completely removed.

## 4. Conclusions

In this study, the applicability of the combined DAF-MF process to respond to changes in raw water quality was demonstrated through a two-year pilot operation. The main findings of this study show that the combined DAF-MF process can successfully respond to changes in raw water quality when optimising the process operating conditions.

When the raw water turbidity was below 10 NTU, the MF-alone process without DAF pre-treatment operated stably, with a TMP increase rate of 0.24 kPa/d, even at a flux of 80 LMH, which is higher than the normal domestic flux of 70 LMH. In this case, a CIP cycle of six months or longer can be suggested.

The TMP increase rate rapidly increased to 43.5 kPa/d at 80 LMH during the period when the raw water turbidity was 10 NTU or higher for a long period of time. In this case, to reduce the TMP increase, the drain-backwash operation was introduced, and the TMC cycle was changed from once per week to once per day. Despite these changes in the operating conditions, the TMP increase rate of the MF-alone process was still relatively high (i.e., 1.41 kP/day). Therefore, the pilot process needed to switch from MF-alone to the combined DAF-MF process. EDS and FTIR analyses confirmed that the TMP increase in the MF-alone process was due to the increase in the presence of algae-related materials and the substantial occurrence of manganese in raw water during the season with high turbidity. Water quality analysis of the chemical cleaning waste solutions obtained after the second CIP also showed that AOM and Mn were the major factors affecting the MF-alone process during this season.

After switching from MF alone to the combined DAF-MF process, the rate of TMP increase during MF in the combined DAF-MF process decreased to 0.64 kPa/d, but this rate was 2.7 times higher than that of the MF-alone process when the raw water turbidity was less than 10 NTU. It was found that the coagulation conditions (coagulant dose and hydraulic surface loading) were unsuitable, generating a large number of small-sized flocs and residual coagulants that accelerated membrane fouling. Particle count analysis for the DAF effluent confirmed that small flocs contributed to membrane fouling. Water quality analysis of the cleaning chemical waste solutions from the third CIP also showed that fine organic flocs and unreacted coagulants not properly separated by the DAF process contributed to the increase in TMP during MF in the combined DAF-MF process. After optimising the DAF process to minimise small flocs and residual coagulants, the rate of TMP increase for the MF process in the combined DAF-MF process significantly decreased and stabilised at 0.17 kPa/d.

Based on the results of this study, the combined DAF-MF process is a suitable approach to respond to changes in the raw water quality by optimising the operating conditions and controlling the factors affecting the process performance. Most lakes and reservoirs in Korea experience algal blooms, high Mn contents, and high turbidity, depending on the season and weather. The combined DAF-MF process can respond well to substantial variations in raw water quality and thus has suitable applicability.

## Figures and Tables

**Figure 1 membranes-11-00964-f001:**
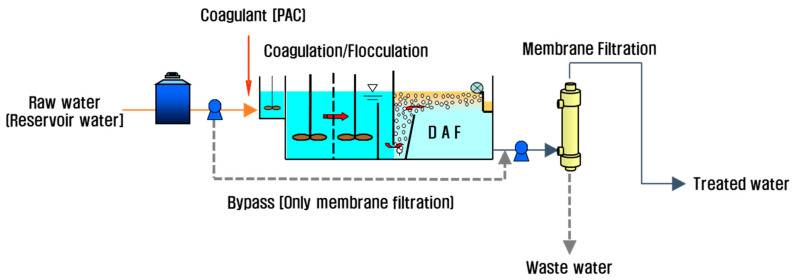
Schematic diagram of the combined DAF-MF process.

**Figure 2 membranes-11-00964-f002:**
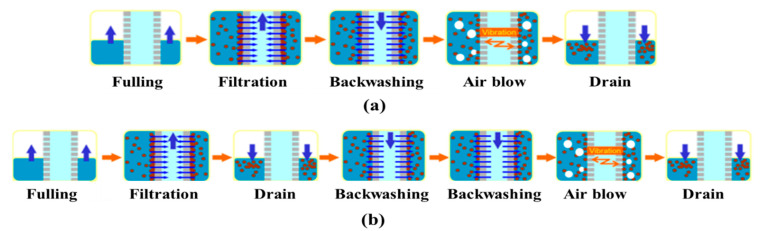
Sequence of (**a**) normal and (**b**) drain-backwash operations of the MF system.

**Figure 3 membranes-11-00964-f003:**
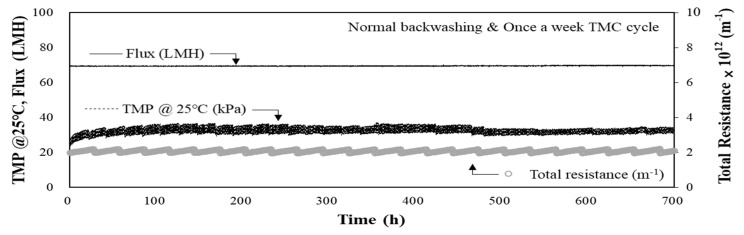
TMP and total resistance during the operation of MF-alone process with low-turbidity raw water (max. turbidity = 11.1 NTU, flux = 70 LMH).

**Figure 4 membranes-11-00964-f004:**
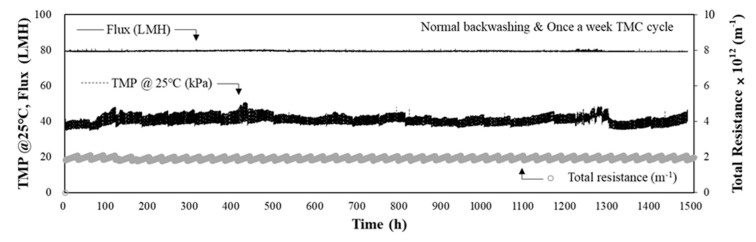
TMP and total resistance during the operation of MF-alone process with low-turbidity raw water (max. turbidity = 13.4 NTU, flux = 80 LMH).

**Figure 5 membranes-11-00964-f005:**
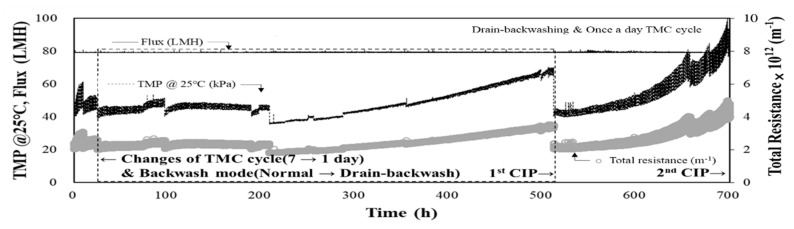
TMP and total resistance during the operation of MF-alone process with high-turbidity raw water (max. turbidity = 76.9 NTU, flux =80 LMH).

**Figure 6 membranes-11-00964-f006:**
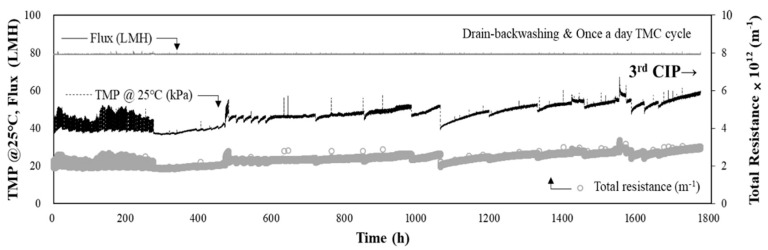
TMP and total resistance during the operation of DAF-MF combined process with high-turbidity raw water (max. turbidity = 81.3 NTU, coagulant dose =60 mg/L, hydraulic loading rate = 15 m/h, flux =80 LMH).

**Figure 7 membranes-11-00964-f007:**
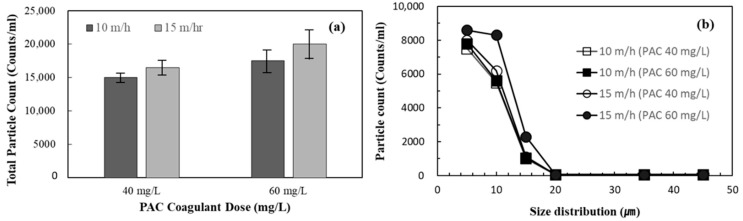
Particle count analysis according to coagulant does and hydraulic loading rate: (**a**) total particle count and (**b**) particle size distribution.

**Figure 8 membranes-11-00964-f008:**
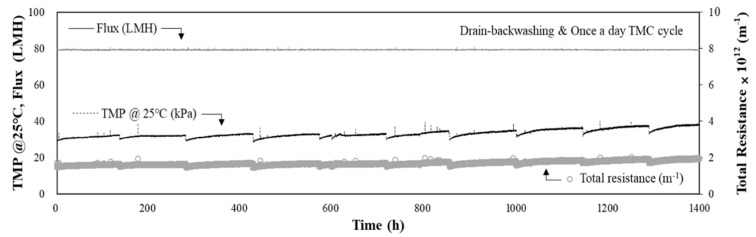
TMP and total resistance during the operation of DAF-MF combined process with high-turbidity raw water (max. turbidity = 83.0 NTU, coagulant dose =40 mg/L, hydraulic loading rate = 10 m/h, flux =80 LMH).

**Figure 9 membranes-11-00964-f009:**
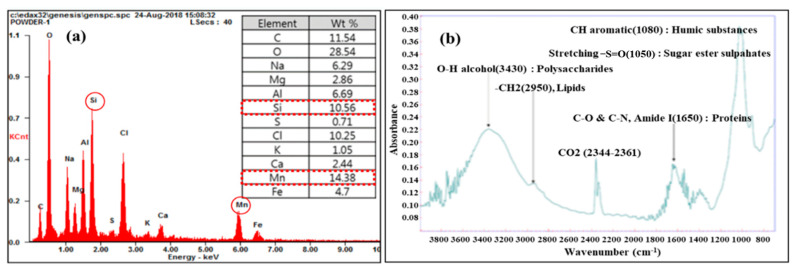
Analysis of raw water quality at the time of rapid TMP increase in the MF-alone process: (**a**) EDS and (**b**) FTIR.

**Table 1 membranes-11-00964-t001:** Raw water (Hoedong Reservoir, Busan) quality.

Water Quality		Membrane Feed Water	
		Raw Water (w/o DAF)	DAF Effluent
Turbidity (NTU)	Average	5.62	<1.0
	Maximum	83.0	2.60
	Minimum	1.35	0.22
DOC (mg/L)	Average	3.06	1.80
	Maximum	4.86	2.67
	Minimum	2.31	1.55
Mn (mg/L)	Average	0.07	0.02
	Maximum	0.24	0.06
	Minimum	0.01	0.01
Chl-a (mg/m^3^)	Average	5.82	0.00
	Maximum	23.5	0.00
	Minimum	0.57	0.00

**Table 2 membranes-11-00964-t002:** Specifications of the membrane module (HFU-2020N, Toray Advanced Materials Korea Inc., Seoul, Korea).

Parameters	Specifications	
Model	HFU-2020N	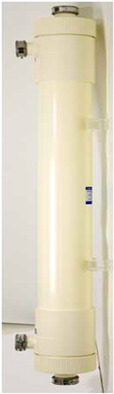
Membrane type	Microfiltration	
Certifications	KWWA F 106	
Module type	Outside-In	
Filtration type	Dead-end/Crossflow	
Permeate turbidity	Less than 0.05 NTU	
Membrane dimension (mm)	In 1.4 / Out 0.8	
Module dimension (mm)	Ø216 X L2,160	
Material	PVDF	
Module material	PVC	
Nominal pore size (μm)	0.01	
Membrane surface area (m^2^)	72	
Max. Operating pressure (kPa)	300	
Max. Operating TMP (kPa)	300	
Max. Temperature (°C)	40	
Adhesive	Polyurethane	
Manufactory	TORAY Advanced Materials Korea Inc.	

**Table 3 membranes-11-00964-t003:** Operating conditions of the MF system in the combined DAF-MF process.

Process	Parameter	Conditions
Filtration/air blowing/backwashing	Flux (LMH)	40~100
	Backwashing (X flow flux, m^3^/m^2^·d)	1.1~1.5
	Air blow flow rate (N m^3^/h)	6~8
	Filtration time (min)	20~60
	Backwashing time (s)	30~60
	Air blow time (s)	30~60
Toray maintenance cleaning (TMC)	NaOCl concentration(mg/L)	300~500
	Submerged time (min)	20~60
	Period cycle (day)	1~7
Clean-in-place (CIP)	1500 mg/L, H_2_SO_4_	2 h
	1500 mg/L, NaOCl	2 h
	3000 mg/L, H_2_SO_4_	2 h

**Table 4 membranes-11-00964-t004:** Analysis of substances eluted during CIP and TMP recovery with each chemical cleaning.

	CIP Condition	TMP Recovery (%)	Al (mg/L)	Mn (mg/L)	Si (mg/L)	Ca (mg/L)	DOC (mg/L)
2nd CIP	3000 mg/L NaOCl	64.6	4.0	3.7	1.1	16	112
	1500 mg/L H_2_SO_4_	2.9	12	9.4	1.2	78	10.8
	1% citric acid	32.1	19	147	0.5	47	-
3rd CIP	1500 mg/L H_2_SO_4_	36.4	42	1.8	15	27	6.5
	3000 mg/L NaOCl	39.7	45	0.8	17	29	91
	1500 mg/L H_2_SO_4_	23.9	2.1	0.29	1.4	4.9	8.3

## Data Availability

The data presented in this study are available on request from the corresponding author.

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
