# Peer review of "Applicability of a Combined DAF-MF Process to Respond to Changes in Reservoir Water Quality through a Two-Year Pilot Plant Operation"

_membranes, 2021, doi:10.3390/membranes11120964_

Round 1

Reviewer 1 Report

The authors investigated the use of DAF-MF combined treatment to address the changes in raw water quality. The authors provided an excellent background to water treatment utilities in Korea and explained the shift from sand-filtration to MF. However, the relevance to the rest of the world is severely lacking. There should be a lot more international references that can be used in this article. DAF-MF is not a new technology in water treatment, although I am not certain about its novelty in its practice in Korea. Based on the introduction, several utilities have already started practicing DAF-MF, hence it can be assumed that pilot studies have already started years ago. The novelty of this work needs to be highlighted. The discussion can be more concise by combining some of the figures (Fig 3 - 6 and 8) into 1 - 2 figures. This will make the discussion less repetitive and the differences will be clearer. 

Below are my specific comments:

Table 1 - are the data provided measured or taken from a source? sample size?

Table 1 - authors should provide the rationale of selecting Mn and Chl-a

Methods - how is TMP measured? not mentioned

Methods - why were different metals measured in raw water and cleaning waste solutions? Again, the basis of selecting the parameters needs to be explained. Is the effluent analyzed?

L41 - these problems - grammar

L155 - TMP cycle - what is TMP cycle? Typo?

L160 - PAC (10%) - 10% refers to?

L177 - MF system recovery is reduced by 2% - what does 'recovery' mean?

Fig 2 - both a and b seemed incorrect.

L197 - degree of membrane fouling is estimated by total resistance, how is the degree of membrane fouling calculated?

Fig 3 - TMP and total resistance are dependent on each other, what is the purpose of showing both parameters?

L386 - source of Si is suggested to be cell wall of diatoms - can suspended particles be a source too?

L399 - CIP? should define when first introduced.

Reviewer 2 Report

The article reports a study evaluating the DAF-MF process for water purification associated with a membrane fouling analysis. The experiments were conducted carefully. The article was generally written well. However, there are still problems being found. Please see comments below.

Major comments:

The results are only from one experimental setup. They are site specific. The authors need to discuss whether or not the results can be applied to other circumstances. This is important for other people who consider using DAF before MF treatment.

Specific comments:

Line 17: Show quantitative information about the water quality before and after deterioration.

Line 51: Are you talking about the plants in Korea? Please make it clear.

Line 55: For MF and UF, give the full names when you show them for the first time.

Lines 72 to 74: Show references and the quantitative results.

Lines 77 to 79: The first four paragraphs show very basic information. If you use them only to justify the necessity of coagulation, flocculation and sedimentation before membrane filtration, you can condense them, because it is common sense.

Line 80: This paragraph needs to be expanded. You started to discuss DAF suddenly. You need to discuss how DAF is better than sedimentation using quantitative information reported by previous studies. Give an overview of DAF. If it is better, why does it not fully replace sedimentation? How does DAF work with membrane?

Lines 85 to 91: Show quantitative information about how DAF outperforms sedimentation.

Lines 95 to 96: I think DAF/membrane has been investigated very well in previous studies in other places of the world. You need to give a literature review.

Line 99: Are there any plants using DAF as the pretreatment? What flux is used with DAF?

Lines 106 to 108: In this paragraph, you discussed sedimentation much more than DAF. Please provide necessary information about DAF pretreatment, such as the flux.

Lines 115 to 116: This is your conclusion, which should not be shown in the introduction.

Line 119: Show background information about CEB and CIP.

Line 136: Why did you analyze manganese specifically? Explain the reason.

Table 3: Please re-design the table to clearly show which parameters correspond to each process. You can show the words on the top, not at the centre of the box.

Figure 2: Figure 2 can be removed. It is not very clear. Please use text to describe the contents.

Line 173: Specify the level.

Line 177: Explain how the MF system recovery was calculated.

Line 182: Specify what EDS and FTIR were used for.

Line 195: Specify the method and the reference.

Line 203: I think you performed CIP many times. Do you mean you calculated TMP recovery for each CIP? Please make this clear.

Line 212: Is this called TMP cycle or TMC cycle?

Lines 213 to 214: Was the membrane new or not? Does this mean the backwashing cannot recover the membrane 100%? I think when the pressure reaches a certain level after backwashing, the membrane cannot be used any more. Please discuss the lifetime of the membrane based on the results.

Line 240: This rate was even less than the previous run with a lower flux. Please explain the reason.

Line 271: Based on the figure, the black and gray lines increased gradually between 200 to 500 h. Why do you say the operation was stable?

Line 272: Please also show the hours besides the days here or in the figure to allow readers to understand this easily.

Line 278: You need to indicate where the results are shown.

Line 288: Why was the increase more rapid? Did CIP make it more rapid? Or was the turbidity in the raw water higher? Please discuss the reason.

Lines 292 to 294: You need to indicate where the results are shown.

Line 310: This paragraph mainly shows the background information to justify the need for optimization. This should be demonstrated in the introduction. Please condense this paragraph.

Lines 337 to 338: Are 14,000 and 20,000 very different? It depends. Show standard deviations.

Lines 345 to 346: You need to indicate where the results are shown.

Figure 7: You should have carried out replicates. Please draw error bars representing standard deviations.

Lines 356 to 358: This is not optimization. You only tried two coagulant doses at two loading rates. You need to perform multiple levels to find the best combination.

Lines 370 to 372: This conclusion is well-known. This is not new. Please highlight what you newly found here.

Line 387: Where was Mn from?

Lines 412 to 414: You need to explain how the CIP was performed in Section 2.

Lines 434 to 436: It is better to perform another CIP after the membrane was run under the optimal condition to see if Al and DOC decrease. A question is that after running for a long time, the membrane is 'dirty'. Cleaning the membrane will result in many substances eluted to the cleaning waste solution. Even if you run the membrane under the optimized condition, you will get the same high concentrations of these substances. To compare the compositions of the cleaning waste solutions before and after optimization, you need to run the membrane for the same length of time and use the same volumes of cleaning solutions.

Round 2

Reviewer 1 Report

The authors have addressed the comments satisfactorily and the paper can now be accepted.

Below are some minor comments to the authors' response:
1. Overall comments: "Until yet," can be deleted.

Reply 1 - "according to standard methods" is way too general. Different countries used different methods while some standard methods do not work under certain conditions.

Reply 4 - were those metals measured representative of membrane fouling?

Reply 6 - 10% volume ratio or mass ratio?